# Prior exposure to *B. pertussis* shapes the mucosal antibody response to acellular pertussis booster vaccination

Evi van Schuppen[1,2,7], Janeri Fröberg [1,2,7], Prashanna Balaji Venkatasubramanian[3], Pauline Versteegen [4], Hans de Graaf[5], Jana Holubová[6], Joshua Gillard[1,2], Pieter G. M. van Gageldonk [4], Irma Joosten[1,2], Ronald de Groot[1,2], Peter Šebo [6], Guy A. M. Berbers[4], Robert C. Read [5], Martijn A. Huynen [3], Marien I. de Jonge [1,2] & Dimitri A. Diavatopoulos [1,2] ✉

*Bordetella pertussis* (Bp), the causative agent of pertussis, continues to circulate despite widespread vaccination programs. An important question is whether and how (sub)clinical infections shape immune memory to Bp, particularly in populations primed with acellular pertussis vaccines (aP). Here, we examine the prevalence of mucosal antibodies against non-vaccine antigens in aP-primed children and adolescents of the BERT study (NCT03697798), using antibody binding to a Bp mutant strain lacking aP antigens (*Bp_mut*). Our study identifies increased levels of mucosal IgG and IgA binding to *Bp_mut* in older aP-primed individuals, suggesting different Bp exposure between aP-primed birth cohorts, in line with pertussis disease incidence data. To examine whether Bp exposure influences vaccination responses, we measured mucosal antibody responses to aP booster vaccination as a secondary study outcome. Although booster vaccination induces significant increases in mucosal antibodies to Bp in both cohorts, the older age group that had higher baseline antibodies to *Bp_mut* shows increased persistence of antibodies after vaccination.

Pertussis, or whooping cough, is a highly contagious acute respiratory disease and one of the least-controlled vaccine-preventable diseases[1]. Several countries that replaced whole cell pertussis (wP) with acellular pertussis (aP) vaccines have experienced an increase in pertussis epidemics in the last decades, with high disease incidence in infants, children and adolescents[2]. Various studies have shown that the first pertussis vaccine given after birth has long-lasting imprinting effects[3–6] that are maintained even after multiple (aP) booster vaccinations given later in life[7–9].

aP vaccines differ both with regard to their antigen content and formulation. They prevent severe disease and mortality due to pertussis in vulnerable infants, but are considered less effective in reducing circulation of *Bordetella pertussis* in the population[7,10–14]. The true prevalence of *B. pertussis* infections is unknown, since the vast majority of individuals infected with *B. pertussis* never undergo diagnostic testing[15,16]. This is particularly the case for infections occurring in vaccinated individuals, which are generally milder or asymptomatic. A better understanding of *B. pertussis* infections and

[1]Laboratory of Medical Immunology, Radboud Institute for Molecular Life Sciences, Radboudumc, 6525 GA Nijmegen, The Netherlands. [2]Radboud Center for Infectious Diseases, Radboudumc, 6525 GA Nijmegen, The Netherlands. [3]Center for Molecular and Biomolecular Informatics, Radboud Institute for Molecular Life Sciences, Radboudumc, 6525 GA Nijmegen, The Netherlands. [4]National Institute for Public Health and the Environment, Centre for Infectious Disease Control, Bilthoven 3720 BA, Netherlands. [5]Faculty of Medicine and Institute for Life Sciences, University of Southampton, Academic Unit of Clinical Experimental Sciences, National Institute of Health Research (NIHR) Clinical Research Facility and NIHR Southampton Biomedical Research Centre, University Hospital Southampton, Southampton, UK. [6]Institute of Microbiology of the Czech Academy of Sciences, Videnska 1083142 20 Prague 4Czech Republic. [7]These authors contributed equally: Evi van Schuppen, Janeri Fröberg. ✉e-mail: Dimitri.diavatopoulos@radboudumc.nl

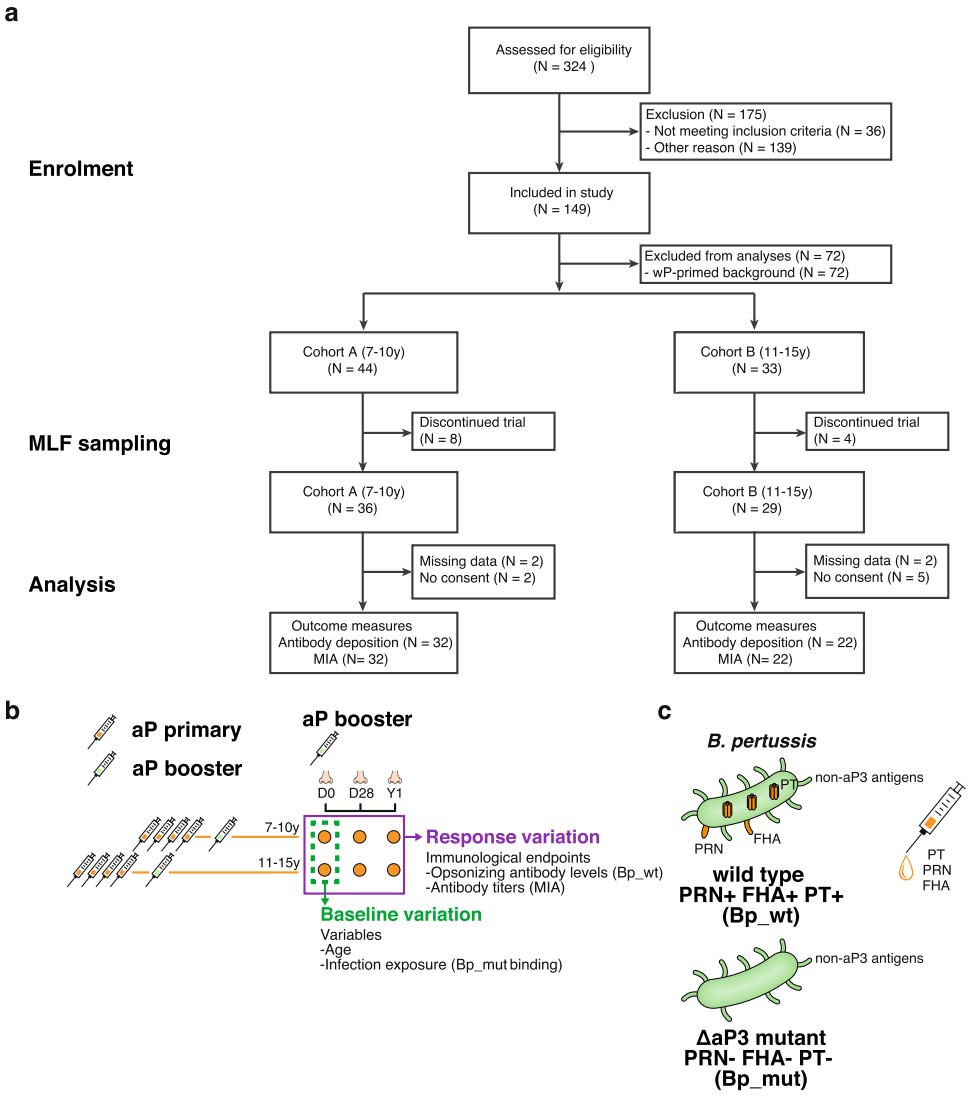

**Fig. 1 | Flow diagram of the booster vaccination study and study procedures.**
**a** Flow diagram describing the recruitment of volunteers, sample sizes, and outcome measures. **b** Overview of the study design, immunological endpoints and framework for analyzing factors contributing to variation in immunological endpoints. Nasal mucosal lining fluid (MLF) was obtained at baseline (D0), and 28 days (D28) and one year (Y1) after vaccination with a dose of the tetanus-diphtheria-acellular pertussis-inactivated polio virus (Tdap-IPV) combination vaccine.
Bacterial antibody deposition and antigen-specific antibody concentrations were used as immunological endpoints to measure response variation. Baseline variation in age and mucosal antibody binding to an isogenic mutant *B. pertussis* strain lacking expression of the aP3 vaccine antigens FHA, PRN, and PT *(Bp_mut)* were included as variables for the response variation. **c** For bacterial antibody deposition experiments, the wild type *B. pertussis* B1917 strain *(Bp_wt)* and *Bp_mut* were used.

their effect on immune memory is important for guiding public health policies.

*B. pertussis* infections are restricted to the respiratory mucosal surfaces, and dissemination beyond the airways is extremely rare[17]. Animal studies have suggested that aP vaccination protects against lower respiratory infection, but is less effective against infection of the upper respiratory tract[18,19]. In this study, we examined the prevalence of nasal antibodies against *B. pertussis* in two aP-primed cohorts, i.e., 7–10 y old and 11–15 y old. Our study identified increased levels of mucosal IgG and IgA binding to *Bp_mut* in older aP-primed individuals, which suggests increased prior exposure to *B. pertussis*. Higher mucosal antibodies at baseline were associated with better persistence of mucosal antibodies at one year after aP booster vaccination. The effect of *B. pertussis* infection on mucosal antibodies was further studied in a controlled human infection model of *B. pertussis*. Taken together, this study provides insight into the prevalence of *B. pertussis* infection-induced mucosal antibodies and their potential effect on the response to aP booster vaccination.

## Results

### Baseline differences in mucosal antibody binding to *Bordetella pertussis*

As part of a larger international multi-center clinical study, i.e., the BERT study, we performed a phase IV open-label longitudinal intervention study in the Netherlands in four different cohorts with varying age and primary pertussis vaccination backgrounds[20]. For this study, we focused on participants primed with aP vaccines during infancy, i.e., children born from 2007 and 2010 (N = 32, cohort A), and adolescents born from 2003 to 2006 (N = 22, cohort B) (Fig. 1a and Table 1). Samples were obtained at baseline as well as 28 days and one year after vaccination with Tdap-IPV (Fig. 1b).

To detect the presence of antibodies against non-aP pertussis antigens, we constructed an isogenic *B. pertussis* mutant strain that lacked the three-component acellular pertussis vaccine (aP3) antigens FHA, PRN, and PT *(Bp_mut)* (Fig. 1c). We used this mutant strain to measure baseline variation in antibodies binding to *B. pertussis*. Figure 2 shows the log2-transformed mean fluorescence intensity

(MFI) of IgM, IgG, and IgA binding to the *Bp_mut* strain in cohorts A and B. IgG and IgA binding to *Bp_mut* was significantly higher in cohort B compared to cohort A ($p = 0.014$ for IgG, and $p = 0.024$ for IgA). Although both three- (aP3) and five-component aP vaccines (aP5) were administered during infancy in both cohorts, aP5 vaccines were more commonly used (Table 1). Given that the *Bp_mut* strain still expresses FIM3, participants who received aP5 vaccines, which includes FIM2/3-antigens, may have residual primary vaccination-induced anti-FIM antibodies that could bind to the *Bp_mut* strain. To examine this further, we stratified each cohort into participants who received either only aP3 vaccines ($N = 2$ for both cohorts), or participants who received one or more aP5 vaccines ($N = 30$ for cohort A and $N = 20$ for cohort B) (Fig. S3). Although the numbers in the aP3-primed group were too low for a formal statistical comparison, antibody binding to the mutant strain at baseline did not seem different between aP3- and aP5-primed

subjects within each respective cohort. These results did not change when we categorized study participants who had received three aP3-vaccine doses and one aP5-vaccine dose as aP3-primed (data not shown). Notably, IgG and IgA binding to *Bp_mut* remained significantly higher in older aP5-primed subjects than in younger aP5-primed subjects. We also examined the correlation between the exact age and baseline antibody binding to *Bp_mut* (Fig. S4), which showed significant positive correlations between age and bacterial binding for all antibody classes, suggesting that older age and not vaccination-induced antibodies to FIM was the major contributing factor to bacterial antibody binding.

## Infection pressure of *B. pertussis*

The increased prevalence of antibody binding to *Bp_mut* suggests increased exposure to *B. pertussis* infection in the adolescent age group. Even though many, if not most *B. pertussis* infections in vaccinated individuals likely remain undiagnosed, we reasoned that these differences may also be reflected in disease notifications. Because pertussis outbreaks are highly cyclical and regional, we evaluated the cumulative pertussis disease incidence in two birth cohorts, representing cohort A (born in 2007–2010) and cohort B (born in 2003–2006), respectively. Pertussis notifications were examined in the same geographical areas where the booster vaccination study was carried out. Pertussis disease incidence was obtained for each year, covering the period from birth until the respective age at inclusion into our study (Fig. 3a). Our analysis indicated different pertussis disease incidence trajectories between the two birth cohorts, with adolescents who were born earlier showing a steeper incline than children who were born later, starting approximately from the age of two to three. When looking at the yearly regional pertussis disease incidence, the difference in trajectories seem to be mainly caused by outbreaks in 2012 and 2014, where the older age group was affected more than the

**Table 1 | Study cohort demographics**

|  | Total *n* = 54 | Cohort A *N* = 32 | Cohort B *N* = 22 |
|---|---|---|---|
| Age, median [IQR] | 8.9 [8.5–12.3] | 8.5 [8.4–8.8] | 12.4 [12.2–12.7] |
| Female sex, *n* (%) | 25 (46) | 18 (56) | 7 (32) |
| **aP priming, *n* (%):** |  |  |  |
| Only aP3 until 11M | 4 (7) | 2 (6) | 2 (9) |
| Only aP5 until 11M | 42 (78) | 29 (91) | 13 (59) |
| One dose of aP5 | 8 (15) | 1 (3) | 7 (32) |
| **aP boosting, *n* (%)** |  |  |  |
| Booster at 4 years | 54 (100) | 32 (100) | 22 (100) |

*aP3* 3 component acellular pertussis, *aP5* 5 component acellular pertussis, *IQR* interquartile range, *11M* 11 months of age (last priming vaccination timepoint).

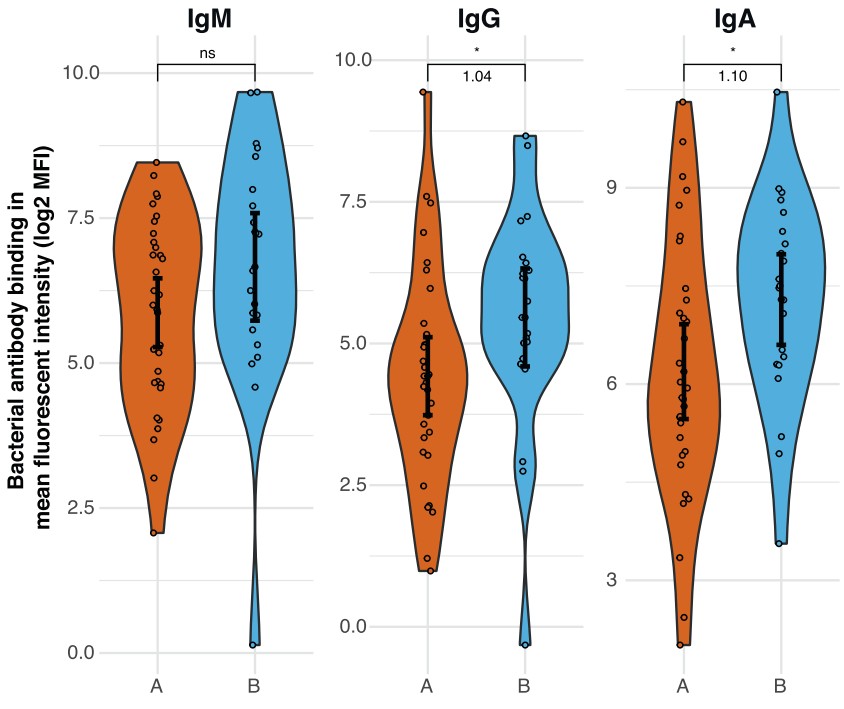

## Cohort

**Fig. 2 | Baseline variation in mucosal antibody binding to *Bp_mut*.** *Bp_mut*, deficient for FHA, PRN, and PT, was incubated with heat-inactivated MLF, after which antibody binding to the bacteria was measured by flow cytometry. Log2-transformed mean fluorescence intensity (MFI) of IgM, IgG and IgA binding to *Bp_mut* in the aP-primed cohorts A and B. Data are $N = 32$ individuals for cohort A

and $N = 22$ individuals for cohort B. Sample means with 95% confidence intervals (solid black point and line) are plotted, and for the significant differences, log2 fold changes are depicted in the figure. Kruskal–Wallis followed by Wilcoxon rank-sum test was used to compare across cohorts. *$p \le 0.05$, $p = 0.014$ for IgG and $p = 0.024$ for IgA.

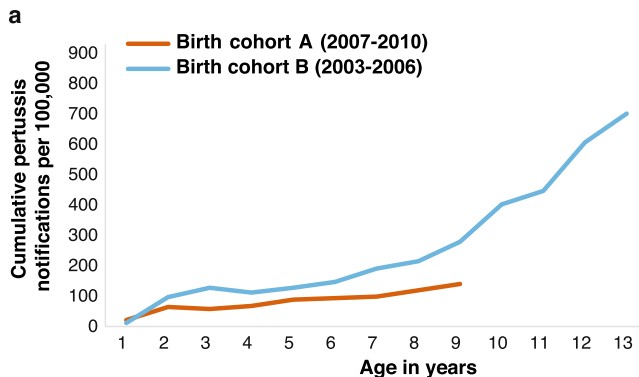

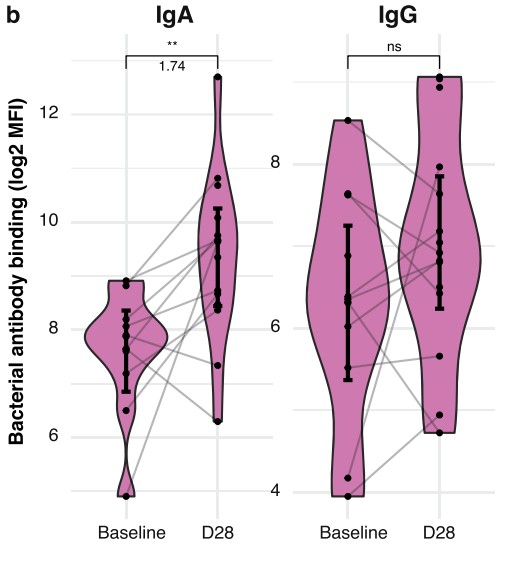

**Fig. 3 | Cumulative *B. pertussis* disease incidence in different aP-primed birth cohorts over time and impact of asymptomatic infection on mucosal antibody responses. a** Cumulative pertussis disease incidence for aP-primed birth cohorts including children born between 2007 and 2010 (red line) and between 2003 and 2006 (blue line), representing cohorts A and B, respectively. Pertussis disease incidence was obtained for postcode-matched regions where the vaccination study was conducted, covering the period from birth until the start of the BERT study. The solid lines reflect the cumulative pertussis disease incidence as the pertussis notifications/100,000 inhabitants. **b** MLF was obtained from adult volunteers prior to (day −7, baseline) and 28 days after intranasal inoculation with *B. pertussis*. *Bp_mut*, deficient for FHA, PRN, and PT, was incubated with heat-inactivated MLF and antibody binding was subsequently measured by flow cytometry. Log2-transformed mean fluorescence intensity (MFI) of IgA and IgG binding to *Bp_mut*. Data are $N = 10$. Sample means with 95% CI (solid black point and line) are plotted. Kruskal–Wallis followed by Wilcoxon signed-rank test was used to test significant differences over baseline in the controlled human infection study. **$p \leq 0.01$, $p = 0.0042$ for IgA.

younger age group (Fig. S5). These data show that different birth cohorts may be exposed to different levels of infection, and that the *Bp_mut* antibody binding assay described in this manuscript may be a valid biomarker for *B. pertussis* infection in aP-primed populations.

### Relationship between *B. pertussis* infection and mucosal bacterial surface-binding antibodies

To establish a direct causal relationship between *B. pertussis* infection in humans and (changes in) mucosal antibody deposition, we made use of the recently established controlled human *B. pertussis* infection model (CHIM)[21]. Previously, in an in-patient pilot study, 15 healthy volunteers were intranasally inoculated with $10^5$ colony forming units

of *B. pertussis*[21]. MLF samples were obtained at baseline as well as 28 days post-challenge, after which antibody deposition was measured as described above. Significant increases in mucosal IgA antibody deposition to *Bp_mut* over baseline were observed at day 28 ($p = 0.004$, Fig. 3b). A similar trend was observed for IgG, although not significant ($p = 0.17$, Fig. 3b). These data suggest that asymptomatic infection with *B. pertussis* increases mucosal antibody binding to *Bp_mut*.

### Mucosal antibody response to Tdap-IPV vaccination in aP-primed children and adolescents

We examined the mucosal antibody response to a dose of Tdap-IPV across the two age cohorts. Samples were collected at day 0, day 28 and one year after vaccination. To assess vaccine immunogenicity, mucosal antibody binding to *Bp_wt* was measured by flow cytometry, and antibody concentrations against the individual aP vaccine antigens were measured by MIA. Binding to *Bp_mut* was included as a negative control (Fig. S6). Both cohorts showed significant increases over baseline with regards to mucosal IgG binding to *Bp_wt* 28 days after vaccination ($p < 0.001$) (Fig. 4a). IgG deposition one year after vaccination remained significantly elevated for cohort B, but not for cohort A (Fig. 4a). IgA showed a non-significant trend towards increased bacterial binding one month after Tdap-IPV vaccination, and waning after one year in both cohorts (Fig. 4b). No significant increases were observed for IgM binding to *Bp_wt* in both cohorts (Fig. S7). Between cohort comparisons per timepoint showed significantly higher IgG and IgA binding levels to *Bp_wt* at all timepoints for cohort B compared to cohort A, but especially for IgG one year after vaccination, suggesting a better IgG persistence in cohort B (Fig. 4c, d).

Using MIA, we also determined the concentration of mucosal IgG and IgA against FHA, PRN, PT and FIM2/3. Significant increases in IgG were observed one month post-vaccination compared to baseline for FHA, PRN and PT (Fig. 5a). The stronger waning of antibody binding to *B. pertussis* seen in younger aP-primed participants (Fig. 4, cohort A) was also observed for IgG concentrations against the individual vaccine antigens, with larger log2 fold reductions between 1 year and baseline in cohort A for all vaccine antigens. None of the cohorts showed a significant mucosal IgA response to the vaccine antigens, although there was a small but significant decrease of PT IgA from day 28 to one year in cohort A (Fig. 5b). As expected, no significant vaccine effect was observed for IgG or IgA against FIM2/3, which was not included in the Tdap-IPV vaccine used in this study. Of note, a small but significant decrease in FIM IgG was observed from day 28 to one year in cohort A (Fig. S8a).

Since we observed higher mucosal IgG *Bp_wt* binding levels 28 days and one year after vaccination in the older aP-primed cohort, we investigated if this was reflected in differences in the response after booster vaccination. We did this by determining the log2-transformed fold changes of IgG and IgA *Bp_wt* binding between day 28 and baseline (FC D28/D0), between one year and day 28 (FC 1Y/D28), and between one year and baseline (FC 1Y/D0). No differences were observed between the responses after booster vaccination in the two aP-primed cohorts, although there was a trend for better persistence after booster vaccination in the older cohort ($p = 0.051$ for IgG FC 1Y/D28 and $p = 0.068$ for IgG FC 1Y/D0, Fig. 6).

### Influence of baseline infection-induced antibodies on the mucosal antibody levels after booster vaccination

Based on the previous analyses, we concluded that there is no difference in the response after booster vaccination between the two cohorts. However, there is a difference in the absolute levels of mucosal Bp-binding antibodies, both at 28 days and one year after vaccination, which are higher in the older aP-primed cohort. To investigate if there was a relation between baseline *Bp_mut* antibody deposition and the subsequent Bp-binding antibodies after

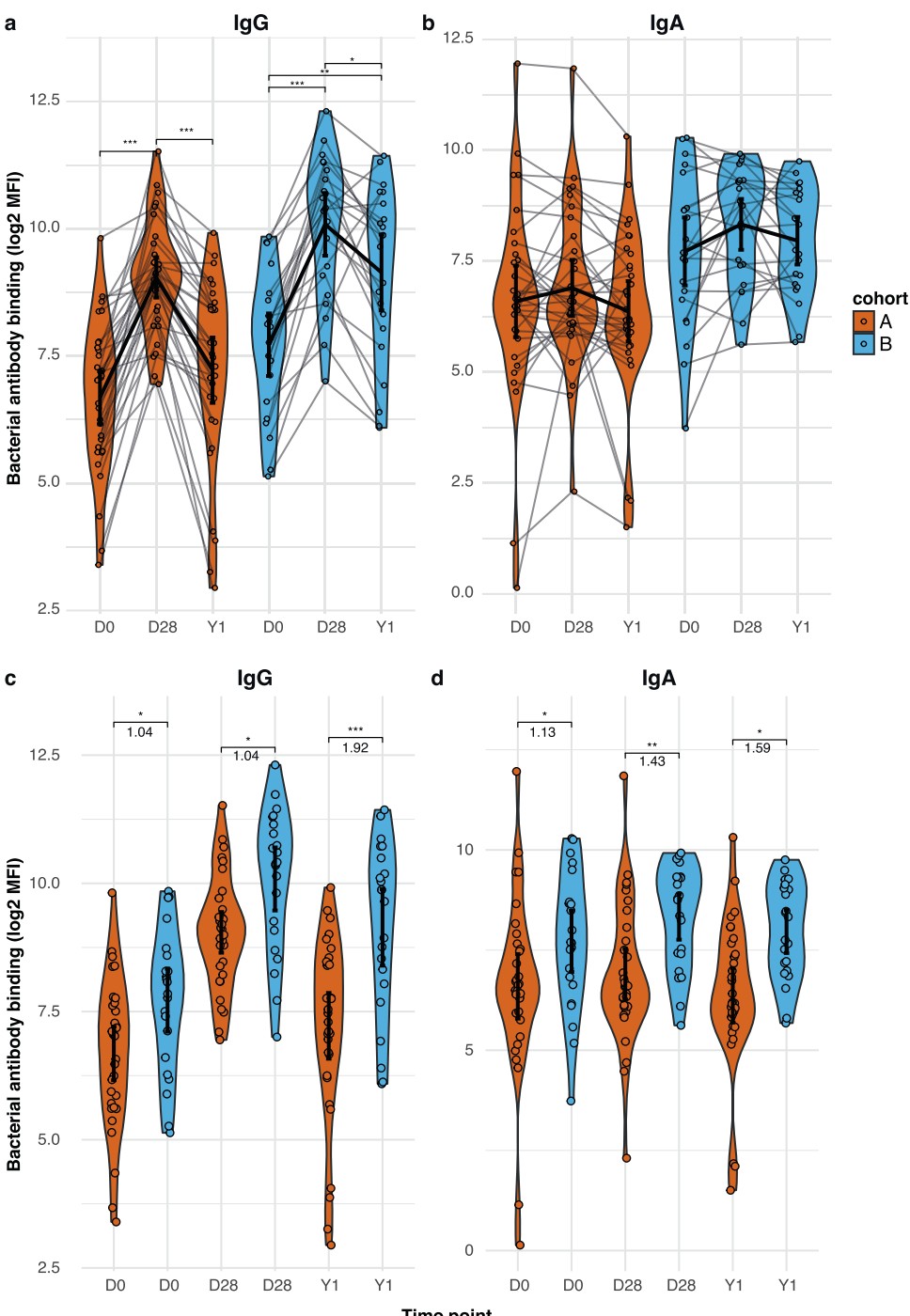

**Fig. 4 | Effect of vaccination on mucosal antibody binding to *B. pertussis*.** MLF was obtained from participants at baseline (D0) and 28 days (D28) and one year (Y1) after a dose of Tdap-IPV. *Bp_wt* was incubated with heat-inactivated MLF and antibody binding to bacteria was subsequently measured by flow cytometry. Log2-transformed mean fluorescence intensity (MFI) of IgG (**a**) and IgA (**b**) binding to *Bp_wt* over time per cohort, and cross-cohort comparison per timepoint for IgG (**c**) and IgA (**d**) binding to *Bp_wt*. Cohort A and cohort B are indicated by color. Data are $N = 32$ individuals for cohort A and $N = 22$ individuals for cohort B. Sample means with 95% confidence intervals (solid black point and line) are plotted. The Friedman with Dunn's post hoc test and Benjamini−Hochberg correction for multiple testing was used to compare timepoints within one cohort (**a** and **b**), and log2 fold change differences can be found in Fig. 6. An unpaired two-sided Wilcoxon rank-sum test was used to compare the levels at each timepoint between the two cohorts (**c** and **d**), and log2 fold change differences between the groups are depicted in the figure. *$p ≤ 0.05$; **$p ≤ 0.01$; ***$p ≤ 0.001$. From left to right exact *p*-values in (**a**) are 4.1e−9, 1.6e−5, 3.0e−6, 2.9e−3, 3.9e−2, in (**c**) are 7.1e−3, 1.5e−2, 1.2e−4, and in (**d**) are 8.1e−3, 4.2e−3, and 5.6e−3.

Tdap-IPV vaccination in aP-primed individuals, we examined the correlations between mucosal baseline IgA and IgG binding levels to *Bp_mut* and mucosal IgG binding levels to *Bp_wt* at day 28 and one year (Fig. 7a, b). We did not stratify on cohort, since the fold changes of the responses did not differ between the cohorts. We excluded mucosal IgA against *Bp_wt*, as there was no vaccine response for this antibody. We observed that higher baseline IgA and IgG antibody deposition on the mutant strain was positively correlated with the IgG antibody levels after aP booster vaccination, at both 28 days and one year post-vaccination.

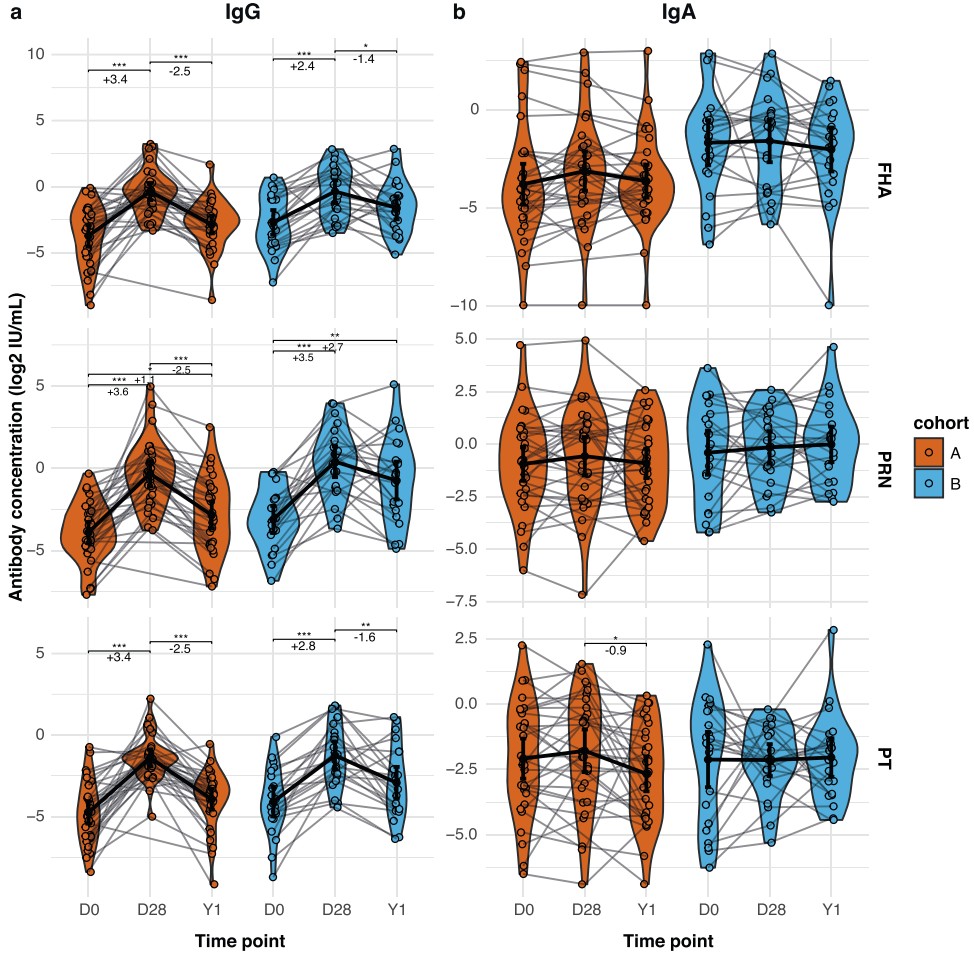

**Fig. 5 | Effect of Tdap-IPV vaccination on aP-antigen-specific mucosal antibody concentrations.** MLF was obtained from participants at baseline (D0) and 28 days (D28) and one year (Y1) after a dose of Tdap-IPV. Antibody concentrations in MLF were measured by multiplex immunoassay (MIA). **a** Log2-transformed concentrations of FHA-, Prn-, and PT-specific IgG in international units (IU)/mL in MLF at the various sample timepoints. **b** Log2-transformed concentrations of FHA-, Prn-, and PT-specific IgA in IU/mL in MLF at the various sample timepoints. Data are $N = 32$ individuals for cohort A and $N = 22$ individuals for cohort B. Sample means with 95% confidence intervals (solid black point and line) are plotted. The Friedman with Dunn's post hoc test and Benjamini–Hochberg correction for multiple testing was used to compare timepoints within one cohort, and log2 fold change differences for the significant changes are depicted in the figure. *$p \leq 0.05$; **$p \leq 0.01$; ***$p \leq 0.001$. From left to right exact $p$-values in (**a**) are 1.3e−8, 3.1e−6, 4.1e−4, and 3.1e−2 for FHA, 5.2e−9, 3.9e−2, 3.3e−5, 2.2e−6, and 1.3e−3 for PRN, and 1.6e−10, 5.9e−7, 2.8e−5, and 5.9e−3 for PT. exact $p$-value in (**b**) is 0.034.

## Discussion

Epidemiological, animal, and modeling studies suggest that continued circulation of *B. pertussis* in vaccinated populations is an important driver of disease and can lead to outbreaks[18,22,23]. Different primary pertussis vaccines induce distinct functional programs in memory T and B cells that can persist for decades[8,24,25]. An important question is how infection influences the memory response to *B. pertussis*, particularly among aP-primed individuals[13]. Since it is difficult to directly detect *B. pertussis* infection by PCR or culture in epidemiological studies, immunological biomarkers of infection may offer a more complete view of *B. pertussis* infections, which may be particularly relevant for aP-vaccinated populations. Here, we developed an innovative immune assay to measure antibody-mediated recognition of an isogenic mutant *B. pertussis* strain that lacks the PRN, PT and FHA antigens that are present in aP3 vaccines. We used this assay to detect bacterial surface-binding antibodies in nasal mucosal lining fluid. This approach provides evidence for increased prevalence of *B. pertussis* infection-induced antibodies among aP-primed adolescents. At this moment, it remains unknown at what age these infections occurred exactly and how long mucosal antibodies to *B. pertussis* persist. A larger follow-up longitudinal epidemiological study is needed to answer these questions.

The differences that we observed between older and younger aP-primed individuals were independent of antibodies against FIM3, which is included in aP5 vaccines. Notably, these age-dependent differences are in line with municipal health surveillance data, indicating that exposure to *B. pertussis* may vary between birth cohorts primed with the same pertussis vaccines.

Boosting with Tdap-IPV increased mucosal IgG deposition on *Bp_wt*, but not IgA. The absence of a consistent increase in mucosal IgA binding after systemic vaccination is in line with other studies demonstrating that IgA responses are primarily induced by infection rather than vaccination[26,27]. Nonetheless, we recently reported significant age-associated increases in FHA-, PRN- and PT-specific IgA in serum after booster vaccination, although much weaker than IgG[20]. Since antibody concentrations in MLF are significantly lower than in serum, sensitivity of detection may play a role. Alternatively, IgA in serum is predominately monomeric, which is not efficiently transported across the mucosal surface by the polymeric Ig receptor[28]. It is also possible that parenteral booster vaccination does not induce production of secretory dimeric IgA antibodies.

Next, we examined whether *B. pertussis* infection-associated mucosal antibodies influence the antibody response to pertussis booster vaccination in aP-primed individuals. We showed that higher

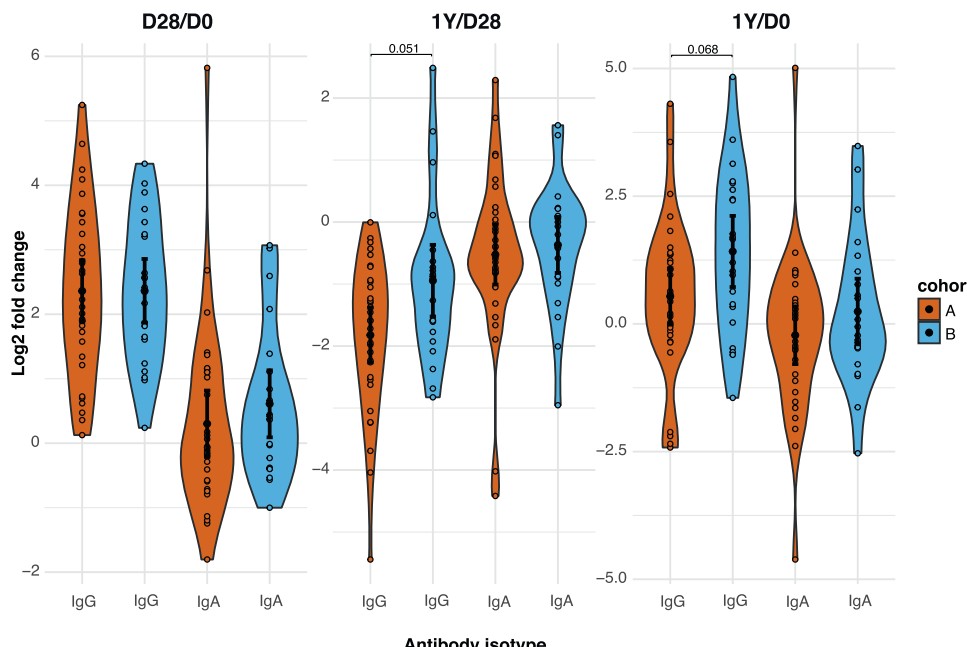

**Fig. 6 | Effect of vaccination on mucosal antibody response to *B. pertussis*.** Longitudinal changes in IgG and IgA *Bp_wt* mucosal deposition levels (log2-transformed mean fluorescence intensity) at D28 over baseline, Y1 over D28 and Y1 over baseline were calculated as log2 fold changes for each cohort. Sample means with 95% confidence intervals (solid black point and line) are plotted. Data are $N = 32$ individuals for cohort A and $N = 22$ individuals for cohort B. An unpaired two-sided Kruskal–Wallis followed by Wilcoxon rank-sum test was used to compare across cohorts.

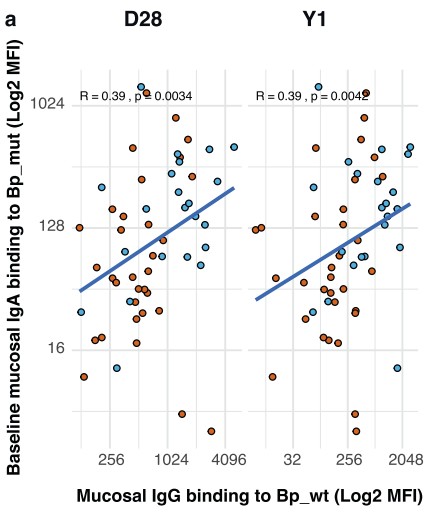

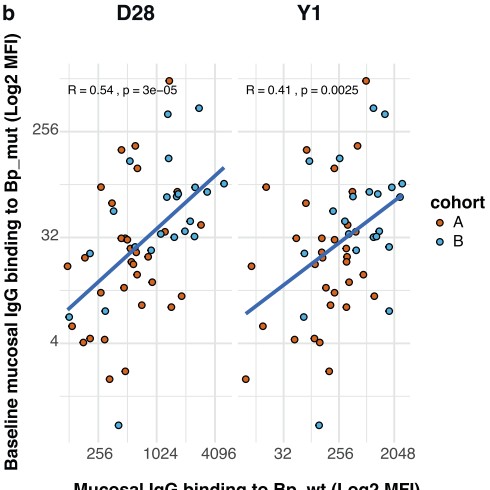

**Fig. 7 | Correlation between baseline *Bp_mut* antibody deposition and the antibody levels after aP booster vaccination in aP-primed individuals.**
**a** Spearman correlation between mucosal IgA binding to *Bp_mut* and mucosal IgG binding to *Bp_wt* at day 28 and one year (log2-transformed mean fluorescence intensity levels). **b** Spearman correlation between mucosal IgG binding to *Bp_mut* and mucosal IgG binding to *Bp_wt* at day 28 and one year (log2-transformed mean fluorenscence intensity levels). For both **a** and **b**, a two-sided spearman correlation was performed and the correlation factor R and *p*-value are depicted in the figure.

baseline antibody binding to the mutant strain is positively correlated with mucosal antibody levels to aP vaccine antigens following vaccination. Mouse studies have shown that, in contrast to wP vaccines, aP vaccines fail to induce tissue-resident memory cells in the upper respiratory tract[29]. However, since the aP vaccine antigens PT, PRN and FHA are also expressed by *B. pertussis* during infection, infection with *B. pertussis* in aP-primed individuals may not only lead to de novo formation of memory B and T cells against non-vaccine antigens, but also activate pre-existing aP antigen-specific memory B and T cells. These aP-antigen-specific memory cells may be re-activated following Tdap-IPV vaccination, which could potentially explain the enhanced

levels and persistence of mucosal antibodies in aP-primed adolescents one month and one year after vaccination.

Our findings suggest that care must be taken when evaluating booster vaccination responses between different cohorts and regions and that infection status should be taken into account along with vaccination status. Improving our understanding of how infection and vaccination continue to shape immune memory is a key area for further research, not only for pertussis but also for other diseases such as COVID-19.

A limitation of our study is that there was limited age variation among the aP-primed participants within each of the cohorts.

Consequently, the effects of infection exposure and age are difficult to separate, and further studies are needed to substantiate these findings. The *Bp_mut* strain is useful to measure the effects of *B. pertussis* infections in aP-primed populations, but is less suited as a biomarker for *B. pertussis* infection in wP-primed subjects as the *Bp_mut* strain still expresses many *B. pertussis* antigens that are also present in wP vaccines. It should be noted that, due to the overlap in antigens expressed by the *Bp_mut* and *Bp_wt* strains, measurements are not independent. This is also observed in Fig. S9a, showing a strong positive correlation between *Bp_mut* and *Bp_wt* IgG binding at baseline. We cannot rule out that exposure to other, non-*Bordetella* bacteria or antigens also results in cross-reactive antibodies that recognize *B. pertussis*. Nonetheless, the results from the controlled human infection study, as well as the absence of increased binding to *Bp_mut* after Tdap booster vaccination, demonstrate that our assay is specific and able to detect transient asymptomatic infections.

Establishing mucosal antibody correlates of protection offers significant advantages over serum-based measures. The highly standardized, well-tolerated and non-invasive mucosal sampling method allows much broader and repetitive sampling of target groups that are normally difficult to sample, including infants and children. Nasal MLF collection in combination with the *Bp_mut* antibody deposition assay allows a quantitative assessment of *B. pertussis* infection exposure in distinct aP-primed populations and geographical regions, independent of variation in (national) disease surveillance systems. This approach may therefore enable evaluation of infection prevalence in the population, not just for *B. pertussis*, but also for other respiratory tract pathogens.

## Methods

### Study population and design

This immunological sub-study is part of a larger international multi-center booster vaccination study (NCT03697798)[20]. For this sub-study, children born between 2007 and 2010 (*N* = 32, 7–10 year old: cohort A), and adolescents born between 2003 and 2006 (*N* = 22, 11–15 year old: cohort B) were included (Fig. 1a). This interventional, longitudinal, open-label phase IV study in four different age cohorts, i.e., 7–10 y (cohort A; aP-primed, *N* = 32), 11–15 y (cohort B; aP- or wP-primed, *N* = 42), 20–34 y (cohort C; wP-primed, *N* = 20), and 60–70 y (cohort D; wP-primed or unvaccinated, *N* = 22), was designed and conducted in accordance with the provisions of the Declaration of Helsinki (1996) and the International Conference on Harmonisation Guidelines for Good Clinical Practice. The trial was registered at the EU Clinical Trial database (EudraCT number 2016-003678-42) and was approved by the Medical Research Ethics Committees United (MEC-U, NL60807.100.17-R17.039) in the Netherlands. Written informed consent was obtained from all participants older than 12 y, as well as from parents or legal guardian of children younger than 16 y, at the start of the study.

This manuscript includes data on the following secondary outcome measures:[20] the concentrations of pertussis toxin (PT) specific IgG antibody one year after vaccination with Tdap-IPV, the change from baseline of antigen-specific IgG antibody levels against other pertussis vaccine antigens (FHA, PRN, FIM2/3) to 28 days and 1 year after vaccination with Tdap-IPV, the change from baseline of functional pertussis-specific antibody levels (i.e., bacterial antibody binding) 28 days and 1 year after vaccination with Tdap-IPV. For this immunological sub-analysis, we focused on aP-primed children born between 2007 and 2010 (*N* = 32, 7–10 year old: cohort A) and aP-primed adolescents born between 2003 and 2006 (*N* = 22, 11–15 year old: cohort B) (Fig. 1a).

The vaccination history of all participants was obtained through the national health institute (RIVM), by reviewing the online records. Children in cohorts A and B received a dose of the diphtheria-tetanus-acellular pertussis-inactivated poliovirus (DTaP-IPV) combination vaccine at 2, 3, 4, and 11 months of age, as recommended by the national immunization program. Each

cohort can be further stratified into participants who exclusively received aP3 vaccine doses (defined as aP3-primed), and participants who exclusively received aP5 vaccine doses (defined as aP5-primed). Participants with a mixed aP3/aP5 background who received at least one dose of aP5 vaccine were classified as aP5-primed. Tdap-IPV booster vaccination was given to all participants in cohorts A and B at the age of 4, according to the Dutch national immunization program. Within this study, all participants received one dose of Tdap-IPV vaccine via intramuscular injection in the upper arm (Boostrix™-IPV - GlaxoSmithKline (GSK), Wavre, Belgium). Boostrix®-IPV contains at least 2 international units (IU) of diphtheria toxoid (DT), at least 20 IU of tetanus toxoid (TT), 8 μg of pertussis toxoid (PT), 8 μg filamentous haemagglutinin (FHA) and 2.5 μg pertussis (PRN). It also contains D-antigen units of different polio virus strains: 40 D-antigen units of type 1 (Mahoney strain), 8 D-antigen units of type 2 (MEF-1 strain), and 32 D-antigen units of type 3 (Saukett strain)[20].

For the controlled human *B. pertussis* infection model (CHIM$_{Bp}$), healthy adult volunteers aged 18–45 years primed with wP during infancy were included[21,30]. This trial was registered with Clinical-Trials.gov (NCT03751514; ethical committee reference 17/SC/0006) and was conducted in accordance with the provisions of the Declaration of Helsinki (1996) and the International Conference on Harmonization Guidelines for Good Clinical Practice. Written informed consent was obtained from all participants. Participants had a history of vaccination against pertussis at least five years before enrollment, were nonsmokers, had no use of antibiotics within four weeks of enrollment, and had no contact with people vulnerable to pertussis disease. Participants with anti-PT IgG concentrations higher than 20 IU/L or a positive culture were excluded. Volunteers challenged with a standard dose of $10^5$ colony forming units (CFU) of *B. pertussis* B1917 were included in the mucosal antibody analysis. These samples were collected from volunteers that received the final inoculum dose that we previously identified as the standard dose[21]. None of the volunteers developed severe symptoms during the follow-up, only mild symptoms were registered.

### Sample collection

Nasal mucosal lining fluid (MLF) samples were obtained from all participants of the vaccination study at baseline, at 28 days (±4 days) after vaccination, and at one year (±4 weeks) after vaccination (Fig. 1b). MLF samples from the volunteers of the CHIM$_{Bp}$ were obtained seven days before (*t* = −7 days) and one month (*t* = 28 days) after challenge. MLF was collected by nasosorption using a synthetic absorptive matrix (SAM, Hunt Developments). SAM strips were gently inserted into the right nostril of the volunteer and placed along the surface of the inferior turbinate. The index finger was lightly pressed to keep the nasosorption device in place and to allow mucosal lining fluid absorption for 60 seconds, after which the nasosorption device was placed back in the protective plastic tube and stored at −80 °C until further analyses.

For elution of MLF, 300 μL of elution buffer (PBS/1% BSA/0.05% Tween20/0.05% azide) was pipetted into a 1.5 mL microcentrifuge tube containing a filter cup with cellulose acetate membrane and placed on ice for 30 minutes to ensure that the filter membrane was blocked to prevent nonspecific protein binding. Following thawing, SAM strips were detached from the holder using sterile forceps by applying pressure at the base of the handle. Thereafter, the SAM strip was placed into the buffer-containing filter in the microcentrifuge tube. Samples were then centrifuged for 20 minutes at 16,000 × *g* at 4 °C. After centrifugation, the filter cup containing the SAM strip was removed, and the eluate was placed into aliquots and transferred to −80 °C until further analyses. All mucosal antibody analyses described were undertaken as exploratory endpoints for both the booster

vaccination study as well as the CHIM$_{Bp}$ and have been approved by the medical ethics committee. The primary outcomes for both studies have already been published[20,21].

## Bacterial strains
*B. pertussis* strain B1917 is a fully genotyped representative of current European isolates[31] and has been used as the challenge strain in the recently developed CHIM$_{Bp}$[30]. Wild type B1917 (*Bp_wt*) expresses the aP antigens filamentous hemagglutinin (FHA), pertactin (PRN), pertussis toxin (PT), and fimbriae 3 (FIM3). For this study, we constructed an isogenic triple mutant in B1917 lacking the *fhaB, prn*, and *ptxS1-S3* genes (B1917 Δ*fhaB* Δ*prn* Δ*ptxS1-S3*), respectively (Table S1) encoding the aP vaccine antigens FHA, PRN, and PT, by allelic exchange using plasmid pSS4245[32,33] (Fig. 1c). Briefly, fragments with ~700 bp length of the upstream and downstream sequences of the respective open reading frames (ORFs) to be deleted were PCR-amplified using appropriate primer pairs (listed in Table S2) and inserted into the corresponding restriction site of pSS4245 used for deletion of the targeted open reading frames by allelic exchange on bacterial chromosome.

The triple mutant strain was sequentially constructed by first deleting *fhaB*, followed by *prn* and finally a *ptxS1-S3* deletion. The B1917 Δ*fhaB* Δ*prn* Δ*ptxS1-S3* is referred to as *Bp_mut*. Of note, the *Bp_mut* strain still expresses FIM3, which is included in some aP vaccine formulations.

The mutations were verified by restriction analysis and resequencing of the corresponding PCR-amplified fragments. Absence of protein products of the *fhaB, prn*, and *ptxS1-S3* genes was confirmed by Western blotting of whole bacterial cell lysates using mouse monoclonal antibodies (MAb) specific for FhaB, for S1 subunit of Ptx (Santa Cruz Biotechnology) and a polyclonal rabbit serum raised against the Prn (data not shown).

Bacteria were harvested at mid-log growth phase (OD$_{620}$ 0.5–0.6) and frozen in aliquots in 15% glycerol at −80 °C until use in the antibody deposition assays below, as described[19].

## Antibody deposition
To measure antibody deposition on the *Bp_wt* and *Bp_mut* strains, $2 \times 10^6$ colony forming units (CFU) of bacteria were incubated with 50% heat-inactivated (30 minutes at 56 °C; to inactivate complement and other inhibitory components) MLF in PBS + 2% BSA for 30 minutes at 37 °C + 5% CO$_2$ while shaking. Subsequently, the bacteria-antibody complexes were centrifuged and fixed in 2% paraformaldehyde for 20 minutes at room temperature. Bacteria were then centrifuged again and resuspended in PBS + 2% BSA containing 1:500 goat polyclonal anti-human IgM-AF647 (Fc-specific, Jackson ImmunoResearch), 1:500 goat polyclonal anti-human IgG-PE (Fc-specific, Jackson ImmunoResearch), and 1:100 goat polyclonal anti-human IgA-FITC (α-specific, Sigma-Aldrich). After 15 minutes incubation at room temperature, surface-bound IgM, IgG, and IgA was measured by flow cytometry on a FACS LSR-II (BD biosciences, SanJose, CA, USA). Heat-inactivated normal human serum (NHS; GTI diagnostics, PHS-N100, lot nr. 2148U) was included as a positive control on each plate (Fig. S1a, b) and gating the wild type and the mutant bacterial populations were based on this condition (Fig. S2a, b). Bacteria alone were measured in all experiments to correct for background antibody binding. The Eight-peak Rainbow bead calibration particles (RCP-30-5A, Spherotech, Lake Forest, IL, USA) were used throughout the study for initial PMT characterization and for setting target Mean Fluorescence Intensities (MFI) values, as well as for daily checks, as described[34]. Data were analyzed using FlowJo Version X (FlowJo, LLC, Ashland, OR, USA) and R.

## Antibody concentrations
Mucosal IgG and IgA concentrations before and after booster vaccination against the individual pertussis antigens FHA, PRN, PT, and FIM2/3 were quantified in independent duplicate using a fluorescent bead-based multiplex immunoassay (MIA), as previously described[35].

Antibodies against FIM2/3, which is not included in the Boostrix-IPV vaccine, were also measured. Analysis was performed with Bio-Plex LX200 in combination with Bio-Plex Manager 6.2 (Bio-Rad Laboratories, Hercules, CA). Pertussis standard serum and control sera were included on each plate. For IgG, the in-house standard used was calibrated against the Pertussis Antiserum (human) 1st WHO International Standard (IS) NIBSC 06/140 and values were assigned in international units (IU/mL) for FHA, PRN, and PT. The in-house standard reference for IgG-FIM2/3 was calibrated against the U.S. Reference Pertussis Antiserum (human) lot 3 and arbitrarily set at 100 AU/mL as previously described[36].

For IgA, the Pertussis Antiserum (human) 1st WHO International Standard (IS) NIBSC 06/140 was used and values were assigned in international units (IU/mL) for FHA, PRN, and PT. Since FIM2/3 IgA concentrations have not been reported in the reference standard, the Pertussis Antiserum (human) 1st WHO IS was arbitrarily set at 100 AU/mL. The LLOD for both IgG and IgA was set at 0.001 IU/mL for FHA, PRN, and PT. For FIM2/3-IgG and FIM2/3-IgA this was 0.001 AU/mL.

## Pertussis disease incidence
Even though pertussis notifications are an underestimation of the true prevalence of *B. pertussis* infections, they may be indicative of the infection pressure. Since pertussis outbreaks are both cyclical and regional, we determined the regional cumulative pertussis disease incidence in the area where the BERT vaccination study was performed. Pertussis disease notifications were obtained for the birth years of participants in cohorts A and B, respectively, covering the period from 2003 until 2017, i.e., the start of the BERT study. Pertussis notifications were obtained from the Dutch National Institute for Public Health and the Environment (RIVM) from the regions Haarlem, Haarlemmermeer, Heemstede, Lisse, Teylingen, and Noordwijkerhout. The number of inhabitants of these regions per year was obtained from the Statistics Netherlands website[37]. Annual pertussis incidence for each birth cohort was calculated by dividing the number of pertussis notifications by the number of inhabitants within the age cohort per year. To estimate the overall pertussis infection pressure in each birth cohort, cumulative pertussis incidences were calculated by adding the disease incidence of the preceding years to each new year.

## Statistical analyses
All statistical analyses were performed using the programming language "R"[38] in the Rstudio environment with libraries 'ggpubr'[39] and 'tidyverse'[40] used for data cleaning and 'ggplot2'[41] used for plotting. MFIs of the opsonization assays were log2-transformed to account for a skewed distribution and data were presented as violin plots with geometric mean concentrations (GMCs) with a 95% confidence interval (CI). Anti-FHA, anti-PRN, anti-PT, and anti-FIM2/3 IgG and IgA levels were log2-transformed and presented as geometric mean concentrations (GMCs) with a 95% confidence interval (CI). A pairwise Wilcoxon signed-rank test was used for comparisons within one cohort between two timepoints, while the differences between two cohorts at one timepoint were analyzed by an unpaired Wilcoxon rank-sum test. Differences between more than one group/timepoint were analyzed by the Friedman with Dunn's post hoc test. Significance cut-off was set at *p*-value < 0.05 and all *p*-values in analyses with many comparisons were corrected for multiple testing using the Benjamini–Hochberg method[42].

The *p*-values and R values of correlations were calculated using the 'lm' command in base R[38], and the 'stat_cor' command of the 'ggpubr' package.

## Reporting summary
Further information on research design is available in the Nature Portfolio Reporting Summary linked to this article.

## Data availability

The processed data generated in this study are provided in the Source data file. The raw data are available from the corresponding author upon reasonable request. The raw data are not publicly available due to data and volunteer's privacy laws. Source data are stored on the open access repository Zenodo[43]. Source data are provided with this paper.

## Material availability

The B. *pertussis* strain B1917 is part of the National Collection of Type Cultures managed by the UK Health Security Agency and can be obtained via www.culturecollections.org (NCTC 14665). The Bp_mut strain can be obtained upon reasonable request from the authors.

## Code availability

The R code that supports the findings of this study is available on the open access repository Zenodo[43].

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

## Author contributions

E.v.S., M.d.J., and D.D. conceived the study. P.V., P.v.G., and G.B. coordinated the booster vaccination study. H.d.G. and R.R. coordinated the controlled human infection model. E.v.S. was responsible for sample processing and antibody deposition assays. E.v.S., P.V., P.v.G., and G.B. were responsible for measuring antibody concentrations using Luminex. J.H. and P.S. constructed the *Bp_mut* bacterial strain. E.v.S., J.F., P.B.V., P.V., J.G., P.v.G., G.B., M.H., M.d.J., and D.D. were responsible for the formal analysis. E.v.S., P.B.V., and D.D. wrote the first draft of the manuscript. E.v.S., J.F., P.B.V., PV, H.d.G., J.H., J.G., P.v.G., I.J., R.d.G., P.S., G.B., R.R., M.H., M.d.J., and D.D. reviewed and revised the draft and agreed to the final submission. PERISCOPE has received funding from the Innovative Medicines Initiative 2 Joint Undertaking under grant agreement No. 115910. This Joint Undertaking receives support from the European Union's Horizon 2020 research and innovation programme and EFPIA and BMGF.

## Competing interests

The authors declare no competing interests.
