## [Peer Review File · Nature Communications]

Prior exposure to *B. pertussis* shapes the mucosal antibody response to acellular pertussis booster vaccinationReviewers' Comments:

Reviewer #1:

Remarks to the Author:

MAJOR COMMENTS

1. Supporting Information – Study population: Please clarify how vaccination history of study participants was confirmed – was this based on review of actual records or individual/parental recall?
2. Line 110, Pertussis disease incidence: The authors have already stated in the introduction that the true prevalence is unknown because of limited diagnostic testing, yet part of this study incorporates pertussis notifications – how much value does this therefore add to this study?
3. Bp_mut: This strain does not contain the 3 pertussis antigens for the vaccine used in this study. However, some infant vaccines also contain FIM. So to address the questions related to persistence of immune responses after infant vaccination, can the authors confirm which infant vaccine was received and if these were 3- or 5-component pertussis vaccine?
4. Line 155 – the authors conclude that increased mucosal IgG and IgA is a function of age based on comparison of cohorts A and B. However, no specific demographics of the cohorts are provided. Within the age range, there is no significant gap between cohorts A and B and if by chance most of the cohort A children are in the upper range limit and most of the cohort B children are in the lower limit, the effect of age is less compelling as an explanation for the findings. It would be ideal to see summary statistics of the demographics of the populations, as well as a supplementary analysis of these data using the specific age of the individuals rather than categorizing by cohort.
5. Figure 2 – I am unclear as to the rationale for assessing responses to Bp_mut in the wP primed individuals. For aP-primed individuals the justification for use of the mutant strain is to differentiate infection vs vaccine responses. For wP-primed individuals the mutant strain cannot do this, so I would suggest removing these data from Figure 2 and including them in a supplementary figure.
6. Lines 158-170 and figure 3: can the authors comment on any time-based changes in diagnostic or other approaches to pertussis that may confound any of these analyses? Are there data that can be used in the same age group over the two time periods to confirm that these are real differences in exposure and not related to something else? In combination with point #2 above – I am not sure this part of the analysis adds to this study and has many limitations.
7. Lines 173-174: The hypothesis is that increased Ab binding is caused by prior infection. Alternatively if any of these individuals received the 5-component pertussis vaccine could this represent maturation of the Ab response over time such that the antibodies have increased affinity/avidity? They certainly have the expertise to do so. In addition, the comment that levels of mucosal antibody 28 days post-challenge are similar to the findings in the study cohort seems hard to fully understand given that in the study cohorts many of the prior infections would be remote – so why would antibody binding be similar in these two presumably different time frames?

MINOR COMMENTS

8. Line 40 states “highest disease incidence in infants, children and adolescents” – please clarify which of these is highest incidence and if referring to all cases or severe disease. This also appears to contradict a later statement “The true prevalence of B. pertussis infections is unknown” (line 47).
9. Line 98 and supplementary: Heading refers to antibody opsonisation, but it would be more accurate to refer to this as antibody deposition since no opsonizing function is being measured in this assay as far as I can tell. If this is correct use of this term should be edited throughout the manuscript.
10. Line 210: why was a 2-fold increase used for nasoconversion? Conventionally for seroconversion a 4-fold increase is used. Is 2-fold within the error of the assay? Data should be included in the supplementary.

Reviewer #2:

None

Reviewer #3:

Remarks to the Author:

This paper examines antibody to pertussis before and after Tdap-IPV vaccination in different age groups (adults & children) and antibody before and after pertussis infection in adults. The strengths of this study are examination of mucosal antibody, the controlled human infection study and the use of a mutated B pertussis strain to measure antibody to proteins not in the Boostrix vaccine. (The mutant does have FIM which is contained in some pertussis vaccines.) However, I do have some concerns about the paper.

They find that antibody responses in older acellular pertussis (aP) and whole cell pertussis (wP) primed children seem similar whereas antibody responses in younger aP primed children seem to decline more rapidly. Hence it seems plausible that pertussis infections have abrogated the expected difference between aP and wP in older children. This is not unreasonable. But the study does not actually compare never infected with previously infected children and does not even attempt this classification. Could other explanations explain this observation?

Other Comments:

1) "Importantly, mucosal antibody responses were significantly improved in individuals with evidence of prior infection, suggesting that prior infections influence aP-induced immunological memory." [lines 31-33]. Since the study does not categorize individuals as never infected vs previously infected, I don't think that this sentence is fully justified. Likewise the section "Prior infection with B. pertussis in aP-primed adolescents is associated with higher mucosal antibody levels after booster vaccination" [pages 9-11] is problematic. If the authors are interested in this question, it is unclear why they do not also for example use mucosal IgA to the mutated strain to classify aP primed children into high and low exposure groups and compare vaccine responses between these two groups.

2) It seems like what the authors call 'opsonization' is actually just binding to the whole bacterium. Opsonization implies that enhancement of phagocytosis was measured. But it seems like the assay only measures binding. Even if it is common, I don't think this terminology is justified.

3) "To adjust for annual fluctuations due to outbreaks, a mean cumulative incidence was calculated by taking the mean incidence of two years; the current year and the year preceding that year. Confidence intervals were calculated using the standard deviation of the two years." [lines 119-122]. Confidence intervals are typically a measure of sampling uncertainty (uncertainty because sample sizes are finite rather than infinite), so I don't see how this method is valid. (But it might be okay to omit these confidence intervals if the number of cases in each region is high.) (Also, since cumulative incidence is over many years, I don't see much rationale in using 'mean cumulative incidence'.)

4) "Although no differences were observed for IgM (Figure 2a), IgG and IgA binding to Bp_mut was significantly higher in 11-15y adolescents compared to 7-10y old children (Figure 2b and 2c, $p = 0.04$ for both IgG and IgA)." [lines 147-150]. It is unclear whether this comparison involves only the aP primed children, which would be better, or whether wP primed older children are also included.

5) In many of the charts it is unclear whether log means log₂, ln or log₁₀. Instead of writing log it would be better to write log₂, ln, log₁₀ et cetera.

6) In Figure 3 the lines for colonized and uncolonized are too difficult to distinguish.

7) In Figure 7 it seems like Mean_FC should be some thing else, perhaps Mean_log₂_FC.

8) In the charts it seems like statistical significance is only shown for certain comparisons. For example, in Figure 2B it seems like D_wP is significantly higher than A_aP, but this is not indicated in the expected way. I don't know a good way of showing statistical significance on these types of charts

when the number of possible comparisons is high. But omitting these comparisons without clearer explanation is confusing and potentially misleading.

Reviewers' comments:

Reviewer #1 (Remarks to the Author):

MAJOR COMMENTS

1. Supporting Information – Study population: Please clarify how vaccination history of study participants was confirmed – was this based on review of actual records or individual/parental recall?

The vaccination history of study participants was confirmed based on review of actual records. We have now included a table including the demographic characteristics per age cohort, i.e. number of participants, age, gender, primary pertussis vaccination background. For the acellular pertussis vaccines, we specified if participants received a 3-component (aP3) or 5-component (aP5) vaccine (Table 1).

2. Line 110, Pertussis disease incidence: The authors have already stated in the introduction that the true prevalence is unknown because of limited diagnostic testing, yet part of this study incorporates pertussis notifications – how much value does this therefore add to this study?

Even though disease notifications are indeed an underestimation of the true prevalence of B. pertussis infections, we reasoned that, within a specific geographical region and time period and with no major changes to the disease surveillance system, clinical disease notifications are likely indicative of infection pressure. Although we did not assess this at an individual level, disease incidence does provide an independent assessment of the infection pressure in the specific health municipality regions for the different birth cohorts. We have explained this more clearly in the manuscript, see lines 123-124. In addition, the steeper cumulative pertussis disease incidence trajectory in the older age cohort in Figure 3a suggest that the differences in exposure to B. pertussis between the cohorts may have occurred already earlier in life (Figure 2). As such, we believe that this analysis contributes significantly to the manuscript.

3. Bp_mut: This strain does not contain the 3 pertussis antigens for the vaccine used in this study. However, some infant vaccines also contain FIM. So to address the questions related to persistence of immune responses after infant vaccination, can the authors confirm which infant vaccine was received and if these were 3- or 5-component pertussis vaccine?

This is a very relevant question as most of the participants in our study received a 5-component vaccine. We have now included a table containing the demographic characteristics per age cohort (see point 1 above), including whether participants exclusively received aP3 vaccine doses, or at least one aP5 vaccine dose. Specifically, we compared antibody deposition on the mutant strain at baseline for aP3-primed children in cohort A (n = 2), aP3-primed adolescents in cohort B (n = 2), aP5-primed children in cohort A (n = 30), and aP5-primed adolescents in cohort B (n = 20). In case aP5-induced antibodies against FIM would contribute significantly to binding to the Bp_mut strain, we would expect differences between the aP3- and aP5-vaccinated individuals at baseline. Although the number of aP3-vaccinated participants in both cohorts is too small for a formal statistical analysis, the limited data suggest there are no significant differences between aP3- and aP5-primed individuals in cohort B (see Figure below and Figure S3 in the manuscript). Additionally, if the differences between the older and younger participants would be caused by aP5-induced antibodies against FIM, we would expect a higher signal for Bp_mut binding in cohort A, since these participants received their last vaccine dose more recently than participants in cohort B. However, we see the opposite, suggesting that the higher binding to Bp_mut in cohort B is caused mainly

by increased exposure to *B. pertussis* infection, rather than vaccination-induced antibodies to FIM. We also assessed whether it made a difference if we re-classified the individuals that had received only a single aP5 vaccine dose as aP3-primed. However, this did not affect the results (data not shown).

Figure S3

4. Line 155 – the authors conclude that increased mucosal IgG and IgA is a function of age based on comparison of cohorts A and B. However, no specific demographics of the cohorts are provided. Within the age range, there is no significant gap between cohorts A and B and if by chance most of the cohort A children are in the upper range limit and most of the cohort B children are in the lower limit, the effect of age is less compelling as an explanation for the findings. It would be ideal to see summary statistics of the demographics of the populations, as well as a supplementary analysis of these data using the specific age of the individuals rather than categorizing by cohort.

We thank the reviewer for this suggestion and have now included a table with demographic characteristics for each cohort (see point 1 above and Table 1). This shows that there is a discrete, but small age gap between children in cohort A (median age: 8.5, IQR: 8.4 – 8.8) and adolescents in cohort B (median age: 12.4, IQR: 12.2 – 12.7). We performed additional correlation analysis of antibody binding to Bp_mut with the actual age of the individuals, rather than categorizing by cohort (see Figure below and Figure S4 in the manuscript). We found a significant positive correlation between mutant baseline antibody deposition and the actual age of the participants, e.g. higher antibody deposition in older individuals, which also points to increased exposure with *B. pertussis* in the adolescents. We hope that this study can form the basis of a larger, international epidemiological study in aP-vaccinated individuals, which would help to obtain a more comprehensive global picture of *B. pertussis* circulation in different age groups, geographical regions and vaccination schedules.

Figure S4

5. Figure 2 – I am unclear as to the rationale for assessing responses to Bp_mut in the wP primed individuals. For aP-primed individuals the justification for use of the mutant strain is to differentiate infection vs vaccine responses. For wP-primed individuals the mutant strain cannot do this, so I would suggest removing these data from Figure 2 and including them in a supplementary figure.

We agree with the reviewer that binding to Bp_mut is less informative for determining exposure to B. pertussis in wP-primed individuals. In line with the reviewer's suggestion, we have therefore decided to omit the results of the wP-primed individuals and only include the results of the aP-primed individuals in Figure 2.

6. Lines 158-170 and figure 3: can the authors comment on any time-based changes in diagnostic or other approaches to pertussis that may confound any of these analyses? Are there data that can be used in the same age group over the two time periods to confirm that these are real differences in exposure and not related to something else? In combination with point #2 above – I am not sure this part of the analysis adds to this study and has many limitations.

To the best of our knowledge, there were no significant changes in diagnostic testing for pertussis in the Netherlands in the time period 2002-2017, or with regard to clinical case definition for pertussis. PCR for diagnosis of pertussis was introduced in the Netherlands in 1997 (Loo 2006). Although comparison of pertussis disease incidence between countries is complicated due to differences in disease surveillance systems, we believe it is valid to compare pertussis disease incidence between the two birth cohorts in the same country and municipal health regions over a relatively short period of time. As such, it provides independent evidence that differences in pertussis disease incidence trajectories exist between two birth cohorts, which likely reflects differences in (asymptomatic) circulation of B. pertussis.

7. Lines 173-174: The hypothesis is that increased Ab binding is caused by prior infection. Alternatively if any of these individuals received the 5-component pertussis vaccine could this represent maturation of the Ab response over time such that the antibodies have increased affinity/avidity? They certainly have the expertise to do so. In addition, the comment that levels of mucosal antibody 28 days post-challenge are similar to the findings in the study cohort seems hard to fully understand given that in the study cohorts many of the prior infections would be remote – so why would antibody binding be similar in these two presumably different time frames?

Please see our answer to point 3 above concerning the potential effects of anti-FIM antibodies. When we analyzed the aP5-primed participants, cohort B showed significantly higher binding to Bp_mut than cohort A. Although these differences could theoretically be caused by affinity maturation of antibodies to FIM, we believe this is unlikely as we did not detect significant differences in the FIM antibody concentrations as determined by multiplex immune assay.

We included the controlled human infection study data in this study since this was a means to obtain pre/post infection samples from asymptotically infected volunteers. These results were not included for direct comparison to the BERT study, but were intended to demonstrate that asymptomatic infection with B. pertussis can lead to increased binding of mucosal antibodies to Bp_mut. We removed any direct comparison between the BERT and the controlled human infection study.

MINOR COMMENTS

8. Line 40 states “highest disease incidence in infants, children and adolescents” – please clarify which of these is highest incidence and if referring to all cases or severe disease. This also appears to contradict a later statement “The true prevalence of B. pertussis infections is unknown” (line 47).

We have clarified this in the manuscript by changing highest incidence into high disease incidence, see line 42. It refers to all pertussis cases.

9. Line 98 and supplementary: Heading refers to antibody opsonisation, but it would be more accurate to refer to this as antibody deposition since no opsonizing function is being measured in this assay as far as I can tell. If this is correct use of this term should be edited throughout the manuscript.

We thank the reviewer for this comment and have replaced antibody opsonization with antibody deposition or antibody binding throughout the whole manuscript.

10. Line 210: why was a 2-fold increase used for nasoconversion? Conventionally for seroconversion a 4-fold increase is used. Is 2-fold within the error of the assay? Data should be included in the supplementary.

Based on the reviewers’ feedback, we have substantially amended the manuscript. As a consequence, the nasoconversion analysis is not featured in the manuscript anymore.

Reviewer #3 (Remarks to the Author):

This paper examines antibody to pertussis before and after Tdap-IPV vaccination in different age groups (adults & children) and antibody before and after pertussis infection in adults. The strengths of this study are examination of mucosal antibody, the controlled human infection study and the use of a mutated B pertussis strain to measure antibody to proteins not in the Boostrix vaccine. (The mutant does have FIM which is contained in some pertussis vaccines.) However, I do have some concerns about the paper.

They find that antibody responses in older acellular pertussis (aP) and whole cell pertussis (wP) primed children seem similar whereas antibody responses in younger aP primed children seem to decline more rapidly. Hence it seems plausible that pertussis infections have abrogated the expected difference between aP and wP in older children. This is not unreasonable. But the study does not actually compare never infected with previously infected children and does not even attempt this classification. Could other explanations explain this observation?

The reviewer is correct that we did not directly compare previously infected children with those that were never diagnosed with pertussis. B. pertussis infections cause a spectrum of clinical severity, ranging from asymptomatic to very severe. We assumed that pertussis disease notifications likely capture the more severe forms of pertussis, but probably fails to identify most of the asymptomatic or mild B. pertussis infections. As we were mainly interested in establishing an assay that is able to also capture these subclinical infections, we included data from asymptotically infected volunteers that participated in our controlled human B. pertussis infection study. The results from the controlled human infection study show that mucosal antibody binding to Bp_mut increases after asymptomatic infection, demonstrating that the assay is specific and able to detect transient asymptomatic infections. There may be other reasons for the observed differences, one of which is the presence of anti-FIM antibodies, which we discussed in detail in point 3 above, as well as in the manuscript. Additionally, we cannot rule out that exposure to other Bordetella species or non-Bordetella bacteria/antigens also resulted in cross-reactive antibodies that recognize B. pertussis. Although we are confident that the observed differences are primarily caused by increased exposure to B. pertussis, also supported by the disease incidence trajectories, we have included these critical discussion points in the manuscript (see lines 310-317).

Other Comments:

1) "Importantly, mucosal antibody responses were significantly improved in individuals with evidence of prior infection, suggesting that prior infections influence aP-induced immunological memory." [lines 31-33]. Since the study does not categorize individuals as never infected vs previously infected, I don't think that this sentence is fully justified. Likewise the section "Prior infection with B. pertussis in aP-primed adolescents is associated with higher mucosal antibody levels after booster vaccination" [pages 9-11] is problematic. If the authors are interested in this question, it is unclear why they do not also for example use mucosal IgA to the mutated strain to classify aP primed children into high and low exposure groups and compare vaccine responses between these two groups.

We agree that without clinical confirmation or a positive diagnostic test, we cannot formally define aP-primed participants with high antibody levels to Bp_mut as 'infected'. Regarding the effect of antibodies to Bp_mut on the immune response to vaccination, we have performed additional analyses. Specifically, we correlated baseline IgA and IgG binding to Bp_mut strain to IgG binding to Bp_wt at day 28 and one year (see Figure 7a and 7b in the manuscript). We showed that a higher baseline antibody deposition

against the mutant strain, indicative of B. pertussis infection, is positively correlated with the antibody levels after aP booster vaccination.

2) It seems like what the authors call 'opsonization' is actually just binding to the whole bacterium. Opsonization implies that enhancement of phagocytosis was measured. But it seems like the assay only measures binding. Even if it is common, I don't think this terminology is justified.

This comment is in line with one of the comments of reviewer #1 (see point 9 above). We have changed antibody opsonization in the manuscript to antibody deposition or antibody binding.

3) "To adjust for annual fluctuations due to outbreaks, a mean cumulative incidence was calculated by taking the mean incidence of two years; the current year and the year preceding that year. Confidence intervals were calculated using the standard deviation of the two years." [lines 119-122]. Confidence intervals are typically a measure of sampling uncertainty (uncertainty because sample sizes are finite rather than infinite), so I don't see how this method is valid. (But it might be okay to omit these confidence intervals if the number of cases in each region is high.) (Also, since cumulative incidence is over many years, I don't see much rationale in using 'mean cumulative incidence'.)

The mean cumulative incidence is used to correct for the yearly fluctuations you see in pertussis notifications. This is done because the periods of exposure in the two age cohorts were not the same, and thus did not contain the same outbreaks. Using the cumulative incidence on its own would have made the difference in the age cohorts bigger, but would have reduced the generalizability of the data, as it would have been highly influenced by the regional outbreaks. The 95% confidence interval is shown to visualize these outbreaks, as they results in a higher confidence interval. By using a more generalized method, we might underestimate the differences in our specific cohort. However, because we want to propose the overall hypothesis that within the aP-primed population different birth cohorts or regions may be exposed to different levels of pertussis, we believe this is the proper approach.

4) "Although no differences were observed for IgM (Figure 2a), IgG and IgA binding to Bp_mut was significantly higher in 11-15y adolescents compared to 7-10y old children (Figure 2b and 2c, p = 0.04 for both IgG and IgA)." [lines 147-150]. It is unclear whether this comparison involves only the aP primed children, which would be better, or whether wP primed older children are also included.

Based on the reviewers' feedback, the manuscript is now mainly focused on aP-vaccinated individuals. As such, we now only show the results (and comparisons) of the aP-primed individuals in the main manuscript (Figure 2). Data from wP-primed individuals is not part of the manuscript anymore.

5) In many of the charts it is unclear whether log means log₂, ln or log₁₀. Instead of writing log it would be better to write log₂, ln, log₁₀ et cetera.

We apologize that this was not clear. In all cases, Log₂ was used for showing the data. We have now specified this in all figures and in the manuscript text.

6) In Figure 3 the lines for colonized and uncolonized are too difficult to distinguish.

Based on (unpublished) results from the controlled human infection study, integrating all bacterial load measurements with immune measurements, we found that classification of subjects as colonized and uncolonized does not accurately reflect infection outcome. Because the number of participants in the controlled human infection study (n=15) is too low to perform subgroup analysis, we have therefore decided not to show the outcome of bacterial inoculation.

7) In Figure 7 it seems like Mean_FC should be some thing else, perhaps Mean_log2_FC.

We thank the reviewer for spotting this error on our part, the fold change is indeed calculated by dividing the log2 values of one timepoints with another timepoint (Log2 D28/D0).

8) In the charts it seems like statistical significance is only shown for certain comparisons. For example, in Figure 2B it seems like D_wP is significantly higher than A_aP, but this is not indicated in the expected way. I don't know a good way of showing statistical significance on these types of charts when the number of possible comparisons is high. But omitting these comparisons without clearer explanation is confusing and potentially misleading.

The differences between two cohorts at one timepoint were analyzed by a Kruskal-Wallis test followed by a Wilcoxon rank-sum test. Significance cut-off was set at p-value < 0.05 and the measures were corrected for multiple testing correction using Benjamini-Hochberg method. As already described in the comments to point 4, we felt that the manuscript should focus primarily on aP-primed subjects. As such, we only show the results of the aP-primed individuals in Figure 2, which includes statistical comparisons between aP-primed individuals.

A Kruskal-Wallis test followed by a pairwise Wilcoxon signed-rank test was used for comparisons within one cohort between two timepoints (Figure 3b, Figure 4, Figure 5, Figure S5, Figure S6, and Figure S7). In these figures the focus is on the response kinetics after vaccination, hence we only depicted the comparison within one cohort.

References

Loo, I. H. M. v. (2006). "Kinkhoest." Tijdschrift voor Infectieziekten **1**(2).

Reviewers' Comments:

Reviewer #1:

Remarks to the Author:

Thank you for the opportunity to review this manuscript. Almost all the previous Reviewers' comments appear to have been adequately addressed. My only outstanding minor suggestion would be to briefly mentioned in the Methods and/or Table 1 that the vaccine history was obtained, by review of actual records.

Reviewer #3:

Remarks to the Author:

Review of Schuppen et al Different exposure to *B. pertussis*

This paper shows that mucosal IgA and IgG (to pertussis) are higher in older children compared to younger children.

It also shows that pertussis challenge boosts mucosal IgA with little effect on mucosal IgG whereas aP vaccination gives the opposite pattern. But there is less emphasis on this point.

1. What is the total number of pertussis immunizations in these children? If this is known and equal between the two groups, this should be stated very clearly as it is critical for interpreting the data.

2. Rather than simply indicating statistical significance, it would be helpful to indicate the fold difference (e.g. fold difference in medians) between the two groups (or two time points) being compared.

3. The title and other parts of the paper suggest better persistence in older children. But their statistical analysis (figure 6) suggests similar fold changes in the two age groups. I feel that this wording needs to be modified in order to avoid being misleading.

4. "These data suggest that asymptomatic infection with *B. pertussis* increases mucosal antibody binding to Bp_mut.". What fraction of these individuals were actually asymptomatic (or mildly symptomatic)? This should be mentioned in the paper.

I also have some minor concerns with regards to the statistical analysis.

5. It is not clear how the confidence interval for cumulative incidence was calculated or even what it means (is it for the Dutch population or for the much smaller number enrolled in the study?).

6. Kruskal Wallis followed by Wilcoxon rank sum seems okay, but for two groups it is almost or exactly equivalent to just Wilcoxon rank sum. But Kruskal Wallis followed by Wilcoxon signed rank is a bit odd as the former ignores pairing. Some other test, like perhaps the Friedman test, may be more appropriate.

7. It's unclear how Benjamini Hochberg was applied. Was it applied at once to all the p values calculated? Or to certain sets of p values?

Some other minor comments. In table 1 meaning of 11M is unclear and total of 13 in only aP3 seems wrong. Table S1 seems to suggest 3 mutant strains as opposed to a single triple mutant.

Rebuttal 2

Reviewer #1

1. Thank you for the opportunity to review this manuscript. Almost all the previous Reviewers' comments appear to have been adequately addressed. My only outstanding minor suggestion would be to briefly mention in the Methods and/or Table 1 that the vaccine history was obtained, by review of actual records.

We appreciate the reviewer's feedback. The vaccine history was indeed obtained by review of online records, using the national Praeventis database in which all vaccinations that are part of the National Immunisation Program are registered. This is now mentioned in the supporting information (lines 14-15).

Reviewer #3

1. What is the total number of pertussis immunizations in these children? If this is known and equal between the two groups, this should be stated very clearly as it is critical for interpreting the data.

Prior to inclusion in the study, every participant received exactly 5 doses of acellular pertussis vaccine: at the age of 2, 3, 4 and 11 months, and at 4 years of age (see Table 1). This is identical between the two age groups and is now clearly stated in the supporting information (lines 15-22).

2. Rather than simply indicating statistical significance, it would be helpful to indicate the fold difference (e.g. fold difference in medians) between the two groups (or two time points) being compared.

*We fully agree with the reviewer, and have now added Log₂ fold changes of the mean for the MIA data depicted in Figure 5 for statistically significant differences. Figure 4 shows longitudinal changes in antibody deposition to *B. pertussis*. Since Figure 6 already shows the Log₂ fold differences between the groups, we therefore decided not to include the Log₂-fold changes in Figure 4. However, we did add a mean and SD to Figure 6 to make comparison between the groups easier.*

3. The title and other parts of the paper suggest better persistence in older children. But their statistical analysis (figure 6) suggests similar fold changes in the two age groups. I feel that this wording needs to be modified in order to avoid being misleading.

*Antibody persistence is not a very well defined term. When comparing the 1-year fold changes over 28 days we observed a near-significant trend for better persistence after booster vaccination in the older cohort (see Figure 6). When comparing antibody concentrations at specific time points between the cohorts, the older age group indeed has significantly higher antibody levels, as reflected in total bacterial antibody binding levels (see e.g. Figure 4), but also in the IgG concentrations at 1 year post-vaccination against PRN and FHA ($p=0.006$ and $p=0.033$, respectively, data not shown). As antibody levels are ultimately the key factor for conferring protection rather than fold-changes, we think this is a key difference between the groups that needs to be highlighted. We have changed the title to: 'Prior exposure to *B. pertussis* shapes the mucosal antibody response to acellular pertussis booster vaccination'.*

4. “These data suggest that asymptomatic infection with *B. pertussis* increases mucosal antibody binding to Bp_mut.”. What fraction of these individuals were actually asymptomatic (or mildly symptomatic)? This should be mentioned in the paper.

As shown in our previous publication on the controlled human infection model, inoculation with B. pertussis did not result in moderate or severe disease symptoms, see the figure below (figure not included in the manuscript). Importantly, the occurrence of AEs did not differ between subjects who became culture-positive and those who remained culture-negative during the 14-day follow-up period. We have added a sentence in the supporting information stating this (line 42-43).

I also have some minor concerns with regards to the statistical analysis.

5. It is not clear how the confidence interval for cumulative incidence was calculated or even what it means (is it for the Dutch population or for the much smaller number enrolled in the study?).

We thank the reviewer for this question. The graph shows the ‘mean cumulative pertussis incidence’, where the pertussis incidence for a given year is averaged over that year and the preceding year. This method is a common method to ‘correct’ for possible yearly outbreak fluctuations. In this case, the confidence interval shows the standard deviation of those two years. Thus, if an outbreak occurred in one of those two years, the CI will be high. We selected this method to reduce the impact of yearly outbreaks and have a more generalized figure, but still be transparent about the effect of outbreak

fluctuations. However, based on the reviewer's comments we realize that this figure may be difficult to interpret. To show changes in the overall exposure to B. pertussis over time in the two birth cohorts, we therefore decided to show the cumulative pertussis incidence, i.e. without averaging to reduce outbreak fluctuations (Figure 3A). For full transparency, we included a supplementary figure to show the pertussis disease incidence rates for each specific year for the two age groups (Figure S5).

6. Kruskal Wallis followed by Wilcoxon rank sum seems okay, but for two groups it is almost or exactly equivalent to just Wilcoxon rank sum. But Kruskal Wallis followed by Wilcoxon signed rank is a bit odd as the former ignores pairing. Some other test, like perhaps the Friedman test, may be more appropriate.

We thank the reviewer for pointing this out. We have now used the Friedman test with Dunn's post-hoc test to check significance when comparing more than two timepoints with dependent data, and the Wilcoxon rank-sum test when comparing the two groups (not paired) or the Wilcoxon signed rank for two timepoints within groups (paired). All figures and significance values were changed accordingly, and the methods in the supporting information have been changed (lines 129-132).

7. It's unclear how Benjamini Hochberg was applied. Was it applied at once to all the p values calculated? Or to certain sets of p values?

Benjamini Hochberg correction was applied to all the p-values at once as a post-hoc correction for multiple testing. This has been clarified in the supporting information (lines 132-134)

Some other minor comments. In table 1 meaning of 11M is unclear and total of 13 in only aP3 seems wrong. Table S1 seems to suggest 3 mutant strains as opposed to a single triple mutant.

We thank the reviewer for spotting the mistake in the total column of table 1, this has now been corrected. The meaning of 11M (11 months of age) has been clarified in the table caption.

Regarding the triple mutant strain, mutants were constructed by sequentially introducing the deletions into the genome of B. pertussis strain B1917. We have clarified this by adapting Table S1.